# Survey of Explainable AI Techniques in Healthcare

**DOI:** 10.3390/s23020634

**Published:** 2023-01-05

**Authors:** Ahmad Chaddad, Jihao Peng, Jian Xu, Ahmed Bouridane

**Affiliations:** 1School of Artificial Intelligence, Guilin University of Electronic Technology, Jinji Road, Guilin 541004, China; 2The Laboratory for Imagery Vision and Artificial Intelligence, Ecole de Technologie Superieure, 1100 Rue Notre Dame O, Montreal, QC H3C 1K3, Canada; 3Centre for Data Analytics and Cybersecurity, University of Sharjah, Sharjah 27272, United Arab Emirates

**Keywords:** explainable AI, medical imaging, deep learning, radiomics

## Abstract

Artificial intelligence (AI) with deep learning models has been widely applied in numerous domains, including medical imaging and healthcare tasks. In the medical field, any judgment or decision is fraught with risk. A doctor will carefully judge whether a patient is sick before forming a reasonable explanation based on the patient’s symptoms and/or an examination. Therefore, to be a viable and accepted tool, AI needs to mimic human judgment and interpretation skills. Specifically, explainable AI (XAI) aims to explain the information behind the black-box model of deep learning that reveals how the decisions are made. This paper provides a survey of the most recent XAI techniques used in healthcare and related medical imaging applications. We summarize and categorize the XAI types, and highlight the algorithms used to increase interpretability in medical imaging topics. In addition, we focus on the challenging XAI problems in medical applications and provide guidelines to develop better interpretations of deep learning models using XAI concepts in medical image and text analysis. Furthermore, this survey provides future directions to guide developers and researchers for future prospective investigations on clinical topics, particularly on applications with medical imaging.

## 1. Introduction

Currently, artificial intelligence, which is widely applied in several domains, can perform well and quickly. This is the result of the continuous development and optimization of machine learning algorithms to solve many problems, including in the healthcare field, making the use of AI in medical imaging one of the most important scientific interests [1]. However, AI based on deep learning algorithms is not transparent, making clinicians uncertain about the signs of diagnosis. The key question then is how one can provide convincing evidence of the responses. However, there exists a gap between AI models and human understanding, currently known as “black-box” [2] transparency. For this reason, many research works focus on simplifying the AI models for better understanding by clinicians, in order to improve confidence in the use of AI models [3]. For example, the Defense Advanced Research Projects Agency (DARPA) of the United States developed the explainable AI (XAI) model in 2015. Later, in 2021, a trust AI project showed that the XAI can be used in interdisciplinary types of application problems, including psychology, statistics, and computer science, and may provide explanations that increase the trust of users [4].

Typically, XAI is an explainable model providing insights into how the predictions are made to achieve trustworthiness, causality, transferability, confidence, fairness, accessibility, and interactivity [5,6]. For example, as shown in Figure 1, it is strongly recommended to allow the AI model to be understandable for the public when the model outputs a decision. It is noted that the definition of XAI is not clear enough according to [7]. In addition, the two words “explainable” and “interpretable” are associated with XAI terms, by which the black-box models are considered “explainable” when the predictions are considered post hoc methods. An “interpretable” model based on the model itself aims to provide human-understandable outputs as steps [8]. It is also important to note that the definition of explainability depends on the prediction task, as mentioned in [9]. Therefore, the term explainable may be measured based on the target users rather than by uniform standards.

In recent years, research outputs of XAI have significantly increased, especially in medical fields, as illustrated in Figure 2. For example, the development of deep learning (DL) models for healthcare has resulted in advanced performance, such as the U-Net model in image segmentation [11]. Despite this progress, DL models still face challenges in clinical practice. Reasons may be related to the inherent high risks in medical decisions. In this context, patients and clinicians are interested to know more about AI-based decisions. In other words, the AI black box is increasingly being used in a variety of high-risk fields where potentially irrational decisions will lead to serious consequences. Therefore, more investigation on XAI is recommended to provide answers to many clinical questions related to accurate and fast diagnosis.

In short, the primary goal of the XAI model is related to people trusting the AI model. For instance, AI users can be divided into two groups: (1) those who have AI “expertise” and (2) those who do not. The first group relates to experts, algorithm developers, and researchers. They focus more on the AI model itself; they develop new methods to monitor the information flow of an algorithm, and explain and optimize the mechanism of an algorithm. The second group of users is generally made up of domain experts, such as radiologists, and the public at large. The expert clinicians require more explanations about AI models to have a technical understanding. Collaborative work between academic and clinical researchers is recommended.

## 2. XAI Techniques Related to Medical Imaging

To trust AI models, the European Union has proposed seven key requirements, including (1) human agency and oversight; (2) technical robustness and safety; (3) privacy and data governance, (4) transparency; (5) diversity, non-discrimination, and fairness; (6) social and environmental well-being; and (7) accountability [12]. These seven requirements are summarized as follows.

### 2.1. Confidentiality and Privacy

AI systems require updating data in real-time, which is a systemic risk. The black-box nature of AI can cause many security problems, which can come from internal or external sources. For example, these problems may be related to the algorithm itself or external, such as improper use of users and the creation of false datasets by network attacks [13,14]. In [15], the authors identified three types of security risks related to black-box AI: network attacks, system bias, and mismatch attacks. These threats can have serious consequences for medical systems. For these reasons, several studies have investigated and proposed solutions to XAI models in terms of data security [16,17].

### 2.2. Ethics and Responsibilities

The medical field has put forward more requirements for the use of AI and how to clarify the ethics and responsibility of AI has become a challenge. For example, irresponsible AI may lead to a loss of medical staff and patients [18]. Furthermore, it involves the ethical issues of data privacy used with AI models. At present, the inspection and accountability of AI are in the early stages. More details about these problems and the interpretable roles of AI can be found in [19,20]. In addition, the concept of responsible AI is discussed and analyzed to develop notions of responsibility for technological domains [21]. Thus, explainability may be an important condition for clarifying these AI responsibilities.

### 2.3. Bias and Fairness

An AI model is trained by datasets having their own inherent attributes, thereby containing hidden bias. For example, in age and skin-color recognition applications using collected portraits of all ages and races, the AI model showed a preference for light skin and a 45-year-old person [22]. Furthermore, the potential risks in big data algorithms that cannot be ignored allow for increased bias and replication or exacerbate human errors [23]. Notably, the bias may be derived from data, algorithms, and user interactions [24]. This will seriously affect the fairness of AI models, causing different people to get different results. In the case of clinical medicine, different patients may show different symptoms that affect the algorithm. Therefore, it is important to foresee the impact of the AI algorithm’s bias in medical healthcare. This should be carefully investigated when deploying AI in personalized medicine.

## 3. Explainable Artificial Intelligence Techniques

This section provides a brief overview of the categories of XAI that can be used in healthcare. According to the literature published in recent years, there are many criteria used to classify XAI methods [25,26,27]. Figure 3 shows the criteria for classifying XAI methods and the corresponding categories. Based on these categories, it can be summarized that the most commonly used XAI techniques in medical fields are shown in Table 1. In addition, Table 2 reports the recent papers using the XAI method. For readability purposes, we have divided the table into explainable methods, modalities, and explanations of how explainable methods are applied. We have categorized the XAI techniques and explained in detail the methods as follows.

### 3.1. Interpretation Types

Two major types of XAI models are represented by intrinsic and post hoc models. These are two different fields of XAI, which is reflected in the way they operate. Intrinsic is also known as model-based interpretability, where the model itself is interpretable by adjusting the structure and/or components. Post hoc models provide an explanation for a trained model by analyzing the original model and an additional one. Despite the widely used post hoc analysis, its application in the medical field requires more effort to become more practical [68]. The intrinsic and post hoc explanations can be summarized as follows.

#### 3.1.1. Intrinsic Explanation

This type of model is structured to be understandable. In this context, conventional models, such as linear regression models, are relatively simple in structure, though capable of being understandable. For example, a model provides the answer along with the corresponding explanation of the linguistic interpreter [69].

#### 3.1.2. Post Hoc Explanation

Post hoc explanation is related to the interpretable information obtained by external methods when analyzing the model (e.g., a neural network after it has been trained). For example, backpropagation [28], class activation mapping [32], and layer-wise relevance propagation [31] are all post hoc interpretation techniques.

### 3.2. Model Specificity

XAI models can be grouped as model-specific and model-agnostic, depending on whether the method can be used in multiple types of models.

#### 3.2.1. Model-Specific Explanation

A model-specific approach is applied to a certain scope of application. For example, the method needs to use a particular structure or property in the model. Furthermore, all intrinsic models are model-specific because all of these methods require the use of the structure of the model itself [70].

#### 3.2.2. Model-Agnostic Explanation

A model-agnostic approach has no special requirements for the model. Rather, it is used in most XAI models. For example, in perturbation-based methods such as local interpretable model-agnostic explanations (LIME) [34], several outputs are obtained and interpreted by perturbing the model inputs. This may be classified as post hoc type.

### 3.3. Explanation Scopes

XAI may be explained over the entire model or for specific inputs and outputs. In this context, two types of explanation can exist: (1) global explanations and (2) local explanations.

#### 3.3.1. Local Explanation

This type considers the model as a black box, focusing on the local variables that contribute to the decision. This leads to the determination of the features which contribute to the decision-making process. Generally, a local explanation focuses on a single input dataset and the characteristic variables associated with it [71].

#### 3.3.2. Global Explanation

This explanation type is interpreted from the model itself. For example, it explains the contribution that relates to the output by getting an understanding of the interaction mechanism of the model variables. This can be formulated as “How does the model predict?” Interpreting the model by a global method depends on the performance of the model. One can generalize the local interpretation to provide an appropriate global interpretation of the model [71].

### 3.4. Explanation Forms

Explanation forms generated by XAI methods can be divided into three main types: feature-based, textual, and example-based. It is worth noting that some methods can generate multiple types of explanations.

#### 3.4.1. Feature Map

Feature-based explanations present the gradient or hidden feature map values as an approximation of the input importance. They can show which part has the greatest impact on the final output. They are usually represented by the original image with an overlay of a saliency map [28,30].

#### 3.4.2. Textual Explanation

This is a human-comprehensible explanation generated in textual form. Semantic descriptions are used to explain the decision of the model. In an example of image captioning, textual explanations are generated in addition to visual interpretations [72].

#### 3.4.3. Example-Based Explanation

This aims to explain a model by presenting one or more examples similar to the given one. The prototypes consist of the features extracted during network training and are designed as examples [38,73].

## 4. Introduction of the Explainable AI Method: A Brief Overview

As mentioned previously, XAI is widely used in many fields, in particular, medical imaging. In this section, we focus on the importance of XAI in healthcare applications.

### 4.1. Saliency

Saliency directly uses the squared value of the gradient as the importance score of different input features [28]. The input can be graph nodes, edges, or node features. It assumes that the higher gradient value is related to the most important features. Although it is simple and efficient, it has several limitations. For example, it can only reflect the sensitivity between the input and output, which cannot express the importance very accurately. In addition, it has a saturation problem. For example, in regions where the performance model reaches saturation, the change in its output relative to any input change is very small, and the gradient can hardly reflect the degree of input contribution.

Guided backpropagation (BP), whose principle is similar to that of the saliency map, modifies the process of backpropagating the gradient [29]. Since the negative gradients are hard to interpret, guided BP only back-propagates the positive gradients and shears the negative gradients to zero. Therefore, guided BP has the same limitations as saliency maps.

One approach to avoid these limitations is to use layer-wise relevance propagation (LRP) [31] and deep Taylor decomposition (DTD) [74]. LRP and DTD are capable of improving a model’s interpretability. In DTD, neural networks use complex non-linear functions that are represented by a series of simple functions. In LRP, the relevance of each neuron in the network is propagated backward through the network, thereby allowing it to quantify the contribution of each neuron to the final output. There are several rules designed with a specific type of layer in a neural network [31,74]. To combine LRP and DTD, LRP can be thought of as providing the framework for propagating relevance through a network, whereas DTD provides the means for approximating the complex non-linear functions used by the network. LRP and DTD may lead to overcoming the limitations of saliency maps and provide more accurate explanations [75].

### 4.2. Class Activation Mapping

Class activation mapping (CAM) is a visualization tool based on convolutional neural networks [32]. It is capable of distinguishing the focus area of the network by obtaining the weight $W_{k}^{c}$ of the recognized feature image $F_{k}$ in the network, where *k* is the unit of the global average pooling layer and *c* represents the class. The convolution between these two parameters $F_{k}$ and $W_{k}^{c}$ provides the feature map $M_{c}\left(x,y\right)$, where *x* and *y* represent the location (x,y) of the convolutional layer. All types of CAM deformation methods are based on graph activation and weights. However, several methods can be used to obtain the weight value. A good explanation of the CAM model is reported in Algorithm 1. Although this is one of the most commonly used algorithms, it has some challenges. For example, the network’s structure requires more flexibility so that the fully connected layer may adapt to the global average pooling layer. For this reason, a new algorithm known as “Gradient-CAM” is proposed to optimize CAM. It uses gradients to compute weight values [33]. First, the network is propagated forward to obtain the feature layer *A* (e.g., output of the last convolutional layer) and the network predicted value *Y* (e.g., output value before softmax activation). Then, the weights *a* are obtained by computing the backpropagation. Finally, the Grad-CAM matrix *L* may be obtained according to $L_{Grad-CAM}^{c}=\mathrm{ReLU}\left(\sum_{k}a_{k}^{c}A^{k}\right)$.
**Algorithm 1** Class activation mapping.**Require:**
Image $I^{C}\left(H,W\right)$; Network *N*
**Ensure:**
Replace FC layer with average pooling layer in Network *N* **procedure** CAM(I, N)    N(*I*)▹ Input image into network    $W_{k}^{c}\leftarrow \left(w_{1},w_{2},w_{3},...,w_{k}\right)$▹ Get weights from average polling layer    $F_{k}^{c}\leftarrow \left(f_{1}\left(x,y\right),f_{2}\left(x,y\right),f_{3}\left(x,y\right),...,f_{k}\left(x,y\right)\right)$
▹ Feature map of the last convolution layer layer    $M_{c}\left(x,y\right)=\sum_{k}w_{k}^{c}f_{k}\left(x,y\right)$▹ Weighted linear summation    $M_{c}\left(x,y\right)=\frac{1}{HW}\sum_{i}^{H}\sum_{j}^{W}M_{c}\left(x,y\right)$▹ Normalize and up-sample to Network input size    $M_{c}\left(x,y\right)=Relu\left(M_{c}\left(x,y\right)\right)$▹ Final image heat map**end procedure**

### 4.3. Occlusion Sensitivity

When training a neural network for image classification, the aim is to know whether this model can locate the position of the main target in the image. By partially occluding the picture, one can observe the situations of the network in the middle layers and the change in the predicted value after inputting the modified image. This leads to an understanding of why the network makes certain decisions. So far, occlusion sensitivity refers to how the probability of a given prediction changes with the occluded part(s) of the image. The higher the output image value, the greater the decrease in the degree of certainty, indicating that the occlusion area is more important in the decision-making process [30].

### 4.4. Testing with Concept Activation Vectors

Testing with the concept activation vectors (TCAV) is an interpretable method proposed by the Google AI team [40]. Textual concepts are related to an explanation that is simple to understand. In the saliency map, it is not possible to explain the concept of pixels. For this reason, TCAV focuses on capturing high-level concepts in the neural network and attempts to provide a linear transformation from input to concepts using directional derivatives to quantify the importance of user-defined concepts to the classification results. However, this technique requires more investigation to be feasible in medical applications.

### 4.5. Triplet Networks

The triplet network (TN) concept is an example-based framework [39]. For example, the TN training set consists of three samples: the first is randomly chosen from the “Anchor” training set, while the other two samples are randomly chosen from the training set in the same “Positive” and different “Negative” categories. By adjusting the parameters based on the distance between three inputs, the technique aims to bring the Anchor closer to the Positive and away from the Negative. Since labeling is not necessary, this method can be used for unsupervised learning. The technique is able to provide an explanation through the similarity between samples too.

### 4.6. Prototypes

A prototype operation can be used as a method to compare the similarity between the target and a typical sample in a category. This sample is known as “prototype” of its class, and it may be easy to understand and compare for users. For example, each unit in the layer stores a weight vector representing an encoded input. The weight can be seen as the feature part of the object. It is interpretable because it makes decisions based on the weighted similarity score extracted from input and prototypes [73]. An example is the “ProtoPNet” CNN model, which has an additional prototype layer used for image classification. This layer takes the output of the previous convolutional layer as input and learns the class prototypes during the training process [38]. The interpretability of ProtoPNet is described as, “this looks like that”, in image classification tasks.

The explainable deep neural network (xDNN) is another prototype-based network for image classification [5]. Two out of five layers are prototypes: (1) prototype and (2) MegaClouds layers. The prototype layer can extract the data distribution and then form the linguistic logical rules if … then … for the explanations as follows.
(1)$$\mathrm{if}\;\left(I∼{\hat{I}}_{P}\right)\;\mathrm{then}\;\left(class\;c\right)$$
where *I* is the input image and ${\hat{I}}_{P}$ represents the prototype. The prototypes that have the same class label are merged into a MegaCloud *M* in the MegaClouds layer. The final expression is as follows.
(2)$$\mathrm{if}\;\left(x∼M_{1}\right)\;\mathrm{or}\;\left(x∼M_{2}\right)\;\mathrm{or}\;\ldots \;\mathrm{or}\;\left(x∼M_{M}\right)\;\mathrm{then}\;\left(class\;c\right)$$
xDNN was evaluated using multiple datasets, including the COVID-CT dataset [10] and the SARS-CoV-2 CT scan dataset [76]. xDNN achieved slightly lower performance metrics (accuracy, sensitivity, and F1-score) compared to the neighboring aware graph neural network (NAGNN), which uses the post hoc method Grad-CAM to interpret the predictions [53].

### 4.7. Trainable Attention

Trainable attention is defined as a group of techniques that focus on important information content in digital multimedia data (e.g., image, video, audio, and text). For example, the combination of text and image information in a hidden layer may let clinicians focus on the corresponding information between the ROI in an image and electronic health records (EHR) [77]. In addition, a multilayer visual attention mechanism can be used to interpret medical image analysis. For example, two attention layers, one close to the input and the other close to the output, can be used for explaining the attention mechanism between both input and output [78]. Despite the recent works related to this trainable attention, more work using this technique is needed to provide a niche of visual information to clinicians.

### 4.8. Shapley Additive Explanations

The Shapley additive explanation (SHAP), which is also a model using Shapley values [36,79], evaluates the importance of an input feature for the final prediction. This model requires much time to calculate the SHAP values. It may combine with other techniques to accelerate the computation of SHAP values [80]. For example, deep explainer (i.e., deep SHAP) is a fast explainability technique that is considered for models with a neural network-based architecture. Generally, SHAP is used to provide explanations and may be used for many clinical topics [61]. However, SHAP applications are still limited to specific problems.

### 4.9. Local Interpretable Model-Agnostic Explanations

Local interpretable model-agnostic explanation (LIME) is a local model-agnostic method that aims to provide an interpretation of the original model by approximating a new simple model from the predictions of a black-box model locally. The new model is used to interpret the results obtained [34]. This advantage allows LIME to be used with any black-box model to interpret only a single prediction. For example, the approximation model is trained as follows. Given a black-box model and the input data, a perturbation is added to the input data, either by overwriting parts of an image or by removing parts of the words from the text. These new samples are then weighted according to their proximity to the corresponding class, an interpretable model is trained on the obtained dataset, and finally, an explanation is obtained by interpreting the model. The explanation produced by the surrogate model can be expressed through Equation (Equation 3).
(3)$$\mathrm{explain}\left(x\right)=\arg\min_{g\in G}L\left(f,g,\pi_{x}\right)+\Omega \left(g\right)$$
where *g* is a surrogate model in all possible interpretable models and *G* is used to explain the instance *x*. *L* is a fidelity function that measures how close the explanation is to the predictions of the original model. Ω(g) infers the complexity of the model *g*, since we want the surrogate model to be interpretable by humans. It minimizes *L* while keeping model complexity, Ω(g), low.

For example, using the least absolute shrinkage and selection operator (Lasso) as a factor of an interpretable model *g* for text classification, LIME can be expressed by K-LASSO with sparse linear explanations, as illustrated in Algorithm 2. Lasso can select features and regularize the weights of features in a linear model. *K* is the number of features selected via Lasso.
**Algorithm 2** Sparse linear explanation using LIME.**Require:**
Classifier *f*; Features number *K*; Instance *x* to be explained; Similarity kernel $\pi_{x}$
 Z={}
 $X_{N}$ = SAMPLE_AROUND$\left(x^{\prime }\right)$ **for** $x_{i}$ in $X_{N}$ **do**    $Z=Z\cup (x_{i},f\left(x_{i}\right),\pi_{x}\left(x_{i}\right))$ **end for**
 ω = K-Lasso(Z,K) **return** ω▹ Explanation for an individual predict

Another algorithm called “GraphLIME” can be used for graph neural networks for classification applications. It extends LIME to work in a non-linear environment by sampling N-hop network neighbors and using the Hilbert–Schmidt independence criterion Lasso (HSIC Lasso) as surrogate models [35]. The explanations for "GraphLIME" have the same patterns as “LIME”, as expressed in Equation (Equation 4)
(4)$$\mathrm{explain}\left(v\right)=\arg\min_{g\in G}L\left(f,X_{n}\right)$$

The difference is that $X_{n}$ represents the sampling local information matrix of node *v* in a graph.

### 4.10. Image Captioning

Image captioning is an intrinsic explanation method that finally provides interpretations of natural languages [41]. Typically, this approach combines CNNs and long short-term memory networks (LSTM) for encoding the image and text, respectively [81]. These techniques are useful to generate medical reports. For example, “TandemNet” can generate visual interpretations in addition to textual explanation [72]. “TandemNet” is a dual attention model that can effectively combine image and text information, extract useful features, and focus attention for accurate image prediction. Medical Visual-Linguistic BERT, Medical-VLBERT, is another algorithm that has been used to generate medical reports of COVID-19 patients [67]. In this context, a curriculum learning framework, competence-based multimodal curriculum learning (CMCL), was proposed to solve the lack of generation of medical reports [82]. It is worth noting that CMCL mimics the learning process of human doctors from an easy to hard approach by scoring the learning difficulty of the training samples and selecting the appropriate difficulty samples for learning at different training stages of the model.

### 4.11. Recent XAI Methods

Based on the recent literature related to XAI, as reported in Table 2 and Table 3, these XAI models can be summarized as follows.

Grad-CAM: It can highlight important areas of a saliency map. It can verify the model accuracy by comparing the deviation between important areas and the actual situation [54]. Generally, saliency is used for abnormality localization in medical images, and it is useful when the detection and segmentation problems can be localized in the desired output network [19]. It is widely applied to medical images and has become one of the most popular XAI methods.

LIME: a tool for visual explanation. It can be used to predict the local output. For instance, it is used to provide image explanation [83]. For example, in practical applications, disturbed pixels need to be set according to requirements, and low repeatability limits the interpretability of LIME [84].

SHAP: It is an important tool for analyzing the features. It is usually used to extract features and conduct attribution analysis. Although time-consuming, some features may be relevant to explain the model [64].

Trainable attention: It is a mechanism used for image location and segmentation. For example, interpretability remains to be determined in the dual attention-gated deep neural network [58]. In [85], the attention mechanism weights were seen as at best noisy predictors of the relative importance of specific regions of the input sequence, and they should not be treated as justifications for the model’s decisions.

## 5. Making an Explainable Model through Radiomics

Radiomics is a method to extract features from medical images. These features, known as radiomic features, have the ability to reveal tissue patterns. Radiomic features are used as input into predictive models for clinical classifications [86,87,88]. Specifically, radiomics can be seen as a multistep process to complete radiomic analysis: (a) image acquisition, (b) image preprocessing, (c) segmentation/labeling leading to identifying regions of interest (ROI), (d) feature extraction and selection, and (e) building predictive models using machine learning [89]. It should be noted that traditional radiomic models consider manual labeling to segment lesions (ROI); this process requires intensive computation and significant effort from radiologists and oncologists to complete the segmentation. With deep learning models, radiomics, also known as deep radiomics, became more practical and was applied in many medical fields, such as pneumonia recognition [54,90]), survival estimation [91,92,93], and survival prediction [94].

The classification of most medical images is a binary problem (e.g., cancer versus non-cancer) focusing on limited and fixed image features for diagnosis. Thus, it uses saliency maps to highlight the important features. For more complex object classification problems, the network usually requires focusing on more local information. As is known, the detection of disease markers is often expensive, and invasive biopsies require significant analysis and time. Therefore, radiomics is used with a variety of imaging modalities to detect diseases by object detection. In this context, it is desired that the radiomic steps be transparent and explainable. Likewise, DL models for use in medical data analysis should be explainable, as described in [88,90,92,93].

Therefore, there is a need to consider increasing the interpretability of the radiomics diagnosis process. However, radiomics is sensitive to image sampling methods, and different sampling methods affect the sampling characteristics [95]. Improving DL interpretability is critical for the advancement of AI with radiomics. For example, a deep learning predictive model is used for personalized medical treatment [89,92,96]. Despite the wide applications of radiomics and DL models, developing a global explanation model is a massive need for future radiomics with AI.

## 6. Discussion, Challenges, and Prospects

Most of the recently proposed medical imaging works use post-interpretation rather than model-based interpretation (e.g., CAM and GCAM models are widely used). In fact, these works focus on the application of algorithms, and the interpretable methods are used as a supplement to the algorithms. In the absence of systematic development of XAI, it is a trend to use local interpretation methods to explain the cases studied. In the case of CNNs and their use in medical images, a saliency map is a simple tool for obtaining an explanation of the areas of interest of the network [97]. Eight interpretable methods of saliency maps (+ Grad-CAM, guided backpropagation, and guided Grad-CAM) were evaluated [19]. However, the performance on the testing datasets was not competitive [19]. Therefore, several challenges still face the XAI technique. These are summarized as follows.

### 6.1. Human-Centered XAI

As AI models involve social decision making, interaction with these models is becoming more important for many users, especially clinicians. The idea is that the interpretation is accepted as an effective tool for communicating with users/persons and AI models.

In a clinical application scenario, XAI provides explanations for doctors and patients. Radiologists also want to know the opinions of others when using AI tools for diagnosis [98]. Obviously, these interpretable methods are not explained to patients. This shows that doctors cannot clearly diagnose diseases in terms of medical treatment. They need strong evidence or more authoritative answers to verify the AI tools. In this case, if the explanations provided by XAI do not meet the doctor’s expectations, these explanations will not be taken into account or considered. It would be hard and not acceptable for patients to be informed that they have been diagnosed using a computer-based tool. Providing an incomprehensible explanation to the patient undoubtedly decreases the trustworthiness of AI. To achieve this goal, many researchers are investigating the design of more sophisticated interpretation methods; they may require more time to be partially trusted [99].

At present, the academic community is focused on the human-centered development of interpretable technology [100,101,102]. Unfortunately, the human–XAI interaction techniques face many challenges that need massive work to be solved [103,104]. In this context, research on human-centered XAI may consider the following: (1) causality (e.g., providing an understandable chain of causal explanations for users), (2) interactivity (e.g., offer explanations from various perspectives, so that users have the option to choose explanations), and (3) counterfactual explanations to enhance human–computer interactions and produces personalized output with the AI model [105].

### 6.2. AI System Deployment

The use of XAI in clinical decision making gives the models more transparency. However, there are many practical problems related to the speed of operation and implementation [106]. For example, developing, deploying, and applying medical-related AI algorithms involves designers, developers, AI product managers, clinicians, and many other people. This will lead to the development of a new management framework for AI models involving social decision making, and their interaction within the framework to address the desired needs of clinicians and patients alike. The idea is that interpretation is a significant tool for communicating with people and AI models. To achieve this goal, many researchers are looking to design more sophisticated interpretation methods to achieve trustfulness [99]. Currently, the academic community is focused on the human-centered development of interpretable technology [100,101,102]. Unfortunately, human–XAI interaction techniques also face many challenges. We require significant work to design tools and methods to effectively and appropriately apply AI technology in medical and healthcare care settings [103,104]. Currently, many institutions provide training steps to clinicians to explain the basics of AI models. It is also strongly recommended to generate performance metrics of bias and accuracy for the algorithms to increase the trust level; see Model Cards from Google [107], AI Fact Sheets from IBM [108], and Datasheets for datasets from Microsoft [109]. So far, the most widely required criteria of the XAI model are: (1) easy to use for users, (2) validity, (3) robustness, (4) computational cost, (5) the ability to fine-tune, and (6) open-source development [110].

### 6.3. Quality of Explanation

When reviewing and investigating papers using the above methods, it was noticed that the XAI functionalities were not as expected when designed. Researchers will face some problems when they use these XAI methods. An XAI model is evaluated by its ability to provide accurate and understandable explanations for its decisions. One can divide the evaluation methods of XAI into two categories: (1) human-centered and (2) computer-centered [111]. In human-centered XAI, the system produces its explanation and is evaluated by human participants. Their feedback is collected and analyzed. However, it requires domain experts such as clinicians to evaluate the explanation performance, making it is highly cost. In computer-centered XAI, the system uses an algorithm to assess the explanation quality of XAI. Among the popular XAI methods, backpropagation has a high rate of success in detecting and recovering from Trojan attacks, particularly for models with large trigger sizes [112]. For example, CAM is effective in detecting the entire trigger region, but may not always provide the accurate localization of the trigger [112]. LIME has a high rate of success in detecting small triggers [112]. In addition, the limited number of examples shows unstable explanations using LIME and SHAP [113]. The pros and cons of the XAI method are summarized in Table 3. The code for these algorithms is also provided. How to evaluate these methods and their explanations is still a popular research challenge [111,114,115], and further research is needed in this context.

### 6.4. Future Directions of Interpretable Models

Most of the current XAI techniques are post hoc (see Table 2). The most likely reason is their ease of use for target users (clinicians, researchers, etc.), who can adapt their methods to post hoc techniques such as Grad-CAM and LIME. However, it is suggested to develop an interpretable model for any high-risk medical situation [8]. In this context, more clinicians are accepting LIME’s chart interpretation, and their satisfaction rate has reached 78%, but most of the clinicians with low satisfaction gave low scores despite their recognition of the explanations provided by LIME [99]. To obtain explanations from the model itself, a general method aims to calculate the loss for the output feature map of each filter in the convolutional layer [116]. This method leads to recognizing which of the filters are activated during the network prediction. In [117], the authors aimed to obtain an abstract structure of a causal model by training a neural network. In [118], the presented method is self-explaining by its similarity to the prototypes, such as comparing specific cases in the network. It is a new form of exploration to explain a GNN by prototype learning. So far, global explainability is desirable in clinical tasks to achieve trust. More particularly, it is necessary for the actual XAI application to take into account users without the necessary AI training. Manual instructions and other further clarifications are strongly recommended. In addition, as reported in [119], an XAI method may be combined with other techniques such as domain adaptation (DA) and federated learning (FL) to achieve better results.

## 7. Conclusions

This paper has presented several popular XAI methods in terms of their principles and deployment in medical image applications, including their performances. The various algorithms were first classified into numerous distinct categories. In the case of the AI currently applied in the medical imaging fields, a popular XAI related to medical image classifications was discussed. A summary of the applications of the recently proposed XAI approaches to increase the interpretability of their proposed models was also detailed. Furthermore, the need for explainable models for radiomic analysis was also explained and discussed. To conclude, discussion and analysis of the medical requirements of XAI, including its prospects and challenges for further investigation, were also given.

## Figures and Tables

**Figure 1 sensors-23-00634-f001:**
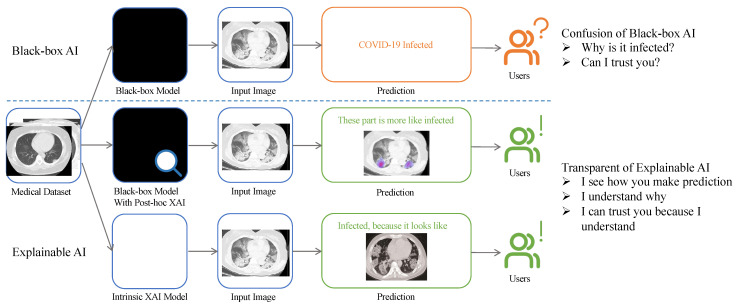
Flowchart of visual comparison between black-box and explainable artificial intelligence, and how the results affect the user. The top branch shows the process of a black-box model. Typically, it provides only results such as classes (e.g., COVID or non-COVID). The other two branches (middle and bottom) represent two XAI methods, referred to in Section 3.1. Specifically, the XAI model (middle) shows the example of saliency map, and the second one (bottom) is the prototype method, as explained in Section 4.1 and Section 4.6, respectively. An example of a CT image obtained from the COVID-19 CT scan dataset [10].

**Figure 2 sensors-23-00634-f002:**
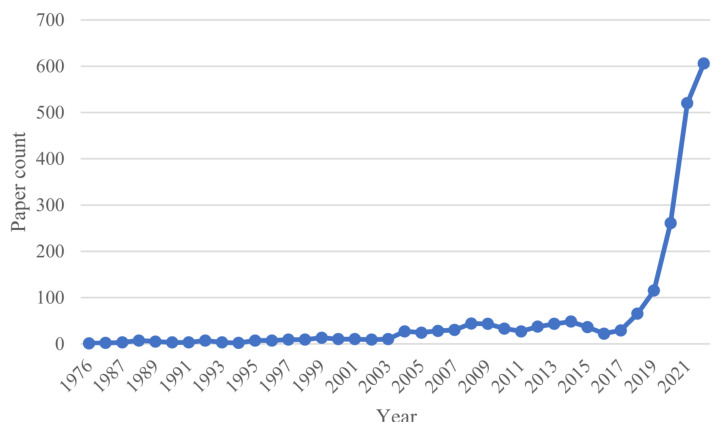
Number of XAI publications added per year from 1976 to 2021. The *x* axis represents the publishing year, and the *y* axis shows the number of publications added in a certain year. The number of publications indexed on PubMed (accessed on 1 July 2022: https://pubmed.ncbi.nlm.nih.gov) that matched the search queries related to explainable AI topics in this survey with a term searched of (explainable AI OR explainable artificial intelligence) AND (medicine OR healthcare).

**Figure 3 sensors-23-00634-f003:**
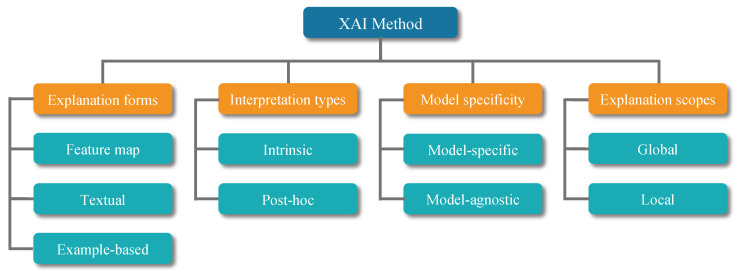
Categorization of explainable AI methods used in this paper. These criteria and categories are summarized from [25,26,27]. The orange and blue grids represent the criteria and categories, respectively.

**Table 1 sensors-23-00634-t001:** Summary of explainable AI techniques classified according to Section 3.

Explanation Type	Paper	Technique	Intrinsic	Post Hoc	Global	Local	Model-Specify	Model-Agnostic
Feature	[28]	BP		*		*	*	
	[29]	Guided-BP		*		*	*	
	[30]	Deconv Network		*		*	*	
	[31]	LRP		*		*	*	
	[32]	CAM		*		*	*	
	[33]	Grad-CAM		*		*	*	
	[34]	LIME		*		*		*
	[35]	GraphLIME		*		*		*
	[36]	SHAP		*		*		*
	[37]	Attention	*			*	*	
Example-based	[38]	ProtoPNet	*			*	*	
	[39]	Triplet Network	*		*	*	*	
	[5]	xDNN	*			*	*	
Textual	[40]	TCAV		*	*	*		*
	[41]	Image Captioning	*			*	*	

“*” indicates it belongs to this category, which is defined in Section 3, BP: backpropagation, CAM: class activation map, LRP: layer-wise relevance propagation, LIME: local interpretable model-agnostic explanations, MuSE: model usability evaluation, SHAP: Shapley additive explanations, xDNN: explainable deep neural network, TCAV: testing with concept activation vectors.

**Table 2 sensors-23-00634-t002:** The following table is a collection of papers that have used interpretable methods in Section 4 to improve the algorithm.

Paper	Organ	XAI	Modality	Contribution
[42]	bone	CAM	X-ray	The model aims to predict the degree of knee damage and pain value through X-ray image.
[43]	lung	CAM	Ultrasound, X-ray	It uses three kinds of lung ultrasound images as datasets, and two networks, VGG-16 and VGG-CAM, to classify three kinds of pneumonia.
[44]	breast	CAM	X-ray	It proposes a globally-aware multiple instance classifier (GMIC) that uses CAM to identify the most informative regions with local and global information.
[45]	lung	CAM	X-ray, CT	The study improves two models, one of them based on MobileNet to classify COVID-19 CXR images, the other one is ResNet for CT image classification.
[46]	lung	CAM	CT	It selects healthy and COVID-19 patient’s data for training DRE-Net model.
[47]	lung	Grad-CAM	CT	It proposes a method of deep feature fusion. It achieves better performance than the single use of CNN.
[48]	chest	Grad-CAM	ultrasound	The paper proposes a semi-supervised model based on attention mechanism and disentangled. It then uses Grad-CAM to improve model’s explainable.
[49]	lung	Grad-CAM	X-ray	It provides a computer-aided detection, which is composed of the Discrimination-DL and the Localization-DL, and uses Grad-CAM to locate abnormal areas in the image.
[50]	colon	Grad-CAM	colonoscopy	The study proposes DenseNet121 to predict if the patient has ulcerative colitis (UC).
[51]	colon	Grad-CAM	whole-slide images	It investigates the potential of a deep learning-based system for automated MSI prediction.
[52]	lung	Grad-CAM	CT	It shows a classifier based on the Res2Net network. The study uses Activation Mapping to increase the interpretability of the overall Joint Classification and Segmentation system.
[53]	chest	Grad-CAM	CT	It proposes a neighboring aware graph neural network (NAGNN) for COVID-19 detection based on chest CT images.
[54]	lung	Grad-CAM, LIME	X-ray	This work provides a COVID-19 X-ray dataset, and proposes a COVID-CXNet based on CheXNet using transfer learning.
[55]	lung	Grad-CAM, LIME	X-ray, CT	It compares five DL models and uses the visualization method to explain NASNetLarge.
[56]	breast	Attention	X-ray	It provides the triple-attention learning $A^{3}$ Net model to diagnose 14 chest diseases.
[57]	bone	Attention	CT	The study introduces a multimodal spatial attention module (MSAM). It uses an attention mechanism to focus on the area of interest.
[58]	colon	Attention	colonoscopy	The proposed Focus U-Net achieves an average DSC and IoU of 87.8% and 80.9%, respectively.
[59]	lung, skin	Saliency	CT, X-ray	The work presents quantitative assessment metrics for saliency XAI. Three different saliency algorithms were evaluated.
[60]	lung	SHAP	EHR	The study introduces a predictive length of stay framework to deal with imbalanced EHR datasets.
[61]	-	SHAP	EHR	The study presents an explainable clinical decision support system (CDSS) to help clinicians identify women at risk for Gestational Diabetes Mellitus (GDM).
[62]	-	SHAP	radiomics	The study proposes a pipeline for interactive medical image analysis via radiomics.
[63]	lung	SHAP	CT	This paper provides a model to predict mutation in patients with non-small cell lung cancer.
[64]	chest	SHAP	EHR	In this paper, it compares the performance of different ML methods (RSFs, SSVMs, and XGB and CPH regression) and uses SHAP value to interpret the models.
[65]	chest	LIME, SHAP	X-ray	The study proposes a unified pipeline to improve explainability for CNN using multiple XAI methods.
[66]	lung	SHAP, LIME, Scoped Rules	EHR	The study provides a comparison among three feature-based XAI techniques on EHR dataset. The results show that the use of these techniques can not replace human experts.
[67]	chest	Image caption	CT	It proposes Medical-VLBERT for COVID-19 CT report generation.

CAM: class activation map, AUC: area under the ROC curve, ROC: receiver operating characteristic curve, LIME: local interpretable model-agnostic explanation, EHR: electronic health record, SHAP: Shapley additive explanations.

**Table 3 sensors-23-00634-t003:** The advantages and disadvantages of the XAI technique. The letters in Code refer to URLs in the footnotes.

Paper	Technique	Simple to Use	Stability	Efficient	Trustworthy	Code	Feature
[33]	Gradient-weighted class activation mapping (Grad-CAM)	+	−	+	−	c1	Works with any CNN
[34]	Local Interpretable Model-agnostic Explanations (LIME)	+	−	−	+	c2	Works on text, image, and tabular data Uses a simple model for an explanation, but complexity must be defined beforehand
[35]	GraphLIME	−	−	−	+	* c3	Works with GNN
[36]	SHapley Additive exPlanations (SHAP)	+	−	−	na.	c4	Have a theoretical foundation from Shapley value
[37]	Trainable attention	−	na.	−	+/−	* c5	Strong anti-noise ability
[5]	xDNN	−	na.	−	+	c6	Features a prototype and Megacloud layer that can effectively extract prototypes
[40]	Testing with Concept Activation Vectors (TCAV)	+	na.	na.	+	c7	Use high-level concept for the explanation
[38]	ProtoPNet	+/−	na.	−	−	c8	Utilized latent space prototypes Have semantic differences between latent space and input, may cause errors in explanation
[41]	Image Caption	+/−	na.	−	+/−	na.	Provide a textual explanation Multiple types of data required

“+” advantage; “−” disadvantage; “*” not official; “na.” unavailable; c1 https://github.com/Cloud-CV/Grad-CAM; c2 https://github.com/marcotcr/lime; c3 https://github.com/WilliamCCHuang/GraphLIME; c4 https://github.com/slundberg/shap; c5 https://github.com/SaoYan/LearnToPayAttention; c6 https://github.com/Plamen-Eduardo/xDNN---Python; c7 https://github.com/tensorflow/tcav; c8 https://github.com/cfchenduke/ProtoPNet. All codes are accessed on 1 September 2022.

## Data Availability

Not applicable.

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
