# Peer review of "Survey of Explainable AI Techniques in Healthcare"

_sensors, 2023, doi:10.3390/s23020634_

Round 1

Reviewer 1 Report

The idea for the paper is very timely and fills an important gap space in the field: the use of AI in biomedical applications is on a trajectory to become ubiquitous, the consequences for failure are significant, and the 'black box' nature of most machine learning algorithms renders the problem a difficult one. The manuscript reviews these issues comprehensively and is both well conceived and well-executed. I found some of the writing a bit off. A good edit for the english, should help.

Author Response

We value the time and effort spent by the Associate Editor and the Reviewers. We are motivated to improve the overall positive received work. Below, we provide our point-by-point responses to the reviewers. We are confident that these changes have resulted in a substantially improved manuscript that addresses the shared concerns of the reviewers. 

 Comments (C_) by reviewers (R1, R2, and R3), and the corresponding answers (A) follow below. 

 Reviewer 1: 

R1C1: The idea for the paper is very timely and fills an important gap space in the field: the use of AI in biomedical applications is on a trajectory to become ubiquitous, the consequences for failure are significant, and the 'black box' nature of most machine learning algorithms renders the problem a difficult one. The manuscript reviews these issues comprehensively and is both well-conceived and well-executed. I found some of the writing a bit off. A good edit for the english, should help. 

 R1A1: We thank the reviewer for pointing out the most relevant aspects to tailor the paper for the readership of AI in biomedical applications. As suggested, we proofread the paper to improve readability and be more precise in our statements.  

Reviewer 2 Report

The authors claim to provide a summary of explainable AI in healthcare. While the authors summarize a significant body of literature and the topic is interesting, the manuscript needs further modification to be publishable.

While the authors claim to provide a summary of explainable AI in healthcare in the title, abstract and background, the later parts of the paper appear to focus almost exclusively on medical imaging or radiology.  I would suggest that the authors reconcile these differences, perhaps by confining the scope of the article to medical imaging systems or radiology. It also appears that many or most of the XAI tools are specific to images. Furthermore, the conclusion does not adequately explain the contribution of the paper.

The authors do not provide sufficient discussion of their figures in the manuscript. With respect to Figure 1, the three branches are not clearly described. With respect to Figure 2, there is no description of where the numbers in this graph came from? If this is a summary of literature searches, what terms were searched, what was done to filter irrelevant or relevant articles in selecting the articles? Which databases were searched? I am also not sure where Figure 3 comes from…why were these categories selected? How does this relate to Table 1? Which came first, was the articles summarized first, which led to the taxonomy in Figure 3 or did Figure 3 come from someone else’s work and then used to create Table 1? I am also not clear on the difference on Tables 1 and 2? Table 2 contains many more papers with little, if any overlap on Table 1…yet they seem to have a lot of similar information.

Early in the paper, the authors describe 2 different user groups.  The first group being the developers and the second being management, general population, etc.  The authors do not seem to include domain experts (e.g., radiologists) who have deep knowledge of the application space but not knowledge of the technology.  It seems that this is a significant oversight. Later in the paper, the authors talk about ease of use. However, it is not clear who the techniques are supposed to be easy to use for…is it the developer, the radiologist, or someone else?

I like the fact that the authors attempt to provide a set of criteria to evaluate the XAI technologies with.  However, this seems rushed and does not have sufficient support. I would suggest that the author either enhance this section or retain the current list but suggest that these are an initial set of criteria but future research should further develop these criteria.

The authors indicate that the user interface may change for the different techniques and this seems apparent as one discusses trainable attention or other areas which might help determine whether the human diagnostician has paid attention to relevant parts of an image. This seems to be another area for potential future work.  This brings the point home that the authors do not provide limitations or future work recommendations in the manuscript.

Detailed Comments:

Line 22 uses the word error. The article is discussing diagnosis and I wonder if error should be diagnosis.

Line 25 appears to end with a period after [3] but then contains a phrase following the .

Line 31, replace “how” with “insight into how”

Line 32 Is XAI only useful in improving trust?

Line 40, what is user level? Does this refer to the user’s knowledge or experience level, or something else?

Section 2.1 is titled “Safety” and yet the entire section discusses cyber security.

Section 2.2 is also titled on one topic but the body of the section seems to have a different topic.

Line 92 Is this a problem with big data or with any algorithm requiring user input for training?

Section 3.2.2 uses the term “state of the art”. Note this is not specific and will change over time. Is there an alternate, more stable description of this class of algorithms?

Line 172. Should the word “light” be “highlight” or some other alternate word?

Line 242. What does lower performance refer to? Is this indicating lower detection performance or lower performance in providing adequate prototypes to support XAI?

Lines 301 through 304 seem to be out of place.  It seems as though the authors might want to include another section which discusses evaluation but it does not seem as though it should be in a section titled image captioning.

While the content of line 339 is correct for diagnosis, diagnosis is only one step in a treatment train and more subtle information is likely needed to formulate treatment.

Line 343, Do biopsies really require computation time? I am not sure what the authors are stating, perhaps that areas which require biopsies might require further analysis which can require significant computation time. Certainly, additional time is needed to perform and analyze a biopsy but I am not sure that this is computational time.

Lines 382 and 383 should simply recommend further research if this is to be achieved…this statement seems judgmental without supporting research.

Table 3, it is not typical for academic research to reference web pages as a primary reference.

Line 402 Ease of use for who, a developer, radiologist, someone else?

Author Response

Kindly find attached a copy of our responses.

Reviewer 3 Report

The paper is well written. It needs more explanation on some techniques.

1. Relevance propagation for neural network must be explained in detail.

2. Taylore decomposition to explain neural network is missing. Please add this.

3. Comparison of performance of famous techniques, especially images classification interpretability. Large amount of work is done to compare various methods in medical imaging. 

Author Response

(The authors gave the same response as above.)

Round 2

Reviewer 3 Report

The paper covers most of the literature on XAI. 

After revision, the quality of the paper has improved.